# Correlation between individual thigh muscle volume and grip strength in relation to sarcopenia with automated muscle segmentation

**Hyeon Su Kim[1], Shinjune Kim[2], Hyunbin Kim[1], Yonghan Cha[3], Jung-Taek Kim[4], Jin-Woo Kim[5], Yong-Chan Ha[6], Jun-Il Yoo [ID][1] ***

1 Department of Orthopedic Surgery, Inha University Hospital, Inha University College of Medicine, Incheon, South Korea, 2 Department of Biomedical Research Institute, Inha University Hospital, Incheon, South Korea, 3 Department of Orthopaedic Surgery, Daejeon Eulji Medical Center, Eulji University School of Medicine, Daejeon, South Korea, 4 Department of Orthopedic Surgery, Ajou University School of Medicine, Ajou Medical Center, Suwon, South Korea, 5 Department of Orthopaedic Surgery, Nowon Eulji Medical Center, Seoul, South Korea, 6 Department of Orthopaedic Surgery, Seoul Bumin Hospital, Seoul, South Korea

* furim@hanmail.net

**Data Availability Statement:** All relevant data for this study are publicly available from the Harvard Dataverse repository (https://dataverse.harvard.

## Abstract

### Introduction

Grip strength serves as a significant marker for diagnosing and assessing sarcopenia, particularly in elderly populations. The study aims to explore the relationship between individual thigh muscle volumes and grip strength, leveraging advanced AI-based UNETR segmentation techniques for accurate muscle volume assessment.

### Methods

The study included 49 participants from a cohort of 478 patients diagnosed with hip fractures at Gyeongsang National University Hospital. We recorded Grip strength and height and utilized UNETR-based segmentation techniques on CT scans, to calculate individual thigh muscle volumes. Point-biserial correlation was employed to explore the relationship between sarcopenia and thigh muscle volumes. The research also included a quantile analysis of grip strength.

### Results

Our findings revealed a strong statistical significance in specific thigh muscles like Rectus femoris, Vastus lateralis, and Vastus intermedius, particularly in males, in relation to sarcopenia. The male cohort displayed a trend where higher thigh muscle volumes correlated with better grip strengths. Meanwhile no such relationship was found within the female group.

edu/dataset.xhtml?persistentId=doi:10.7910/DVN/GH70UC).

**Funding:** The authors received no specific funding for this work.

**Competing interests:** The authors have declared that no competing interests exist.

## Conclusion

The findings indicate that stronger grip strength correlates with larger thigh muscles in males but not in females, with specific muscles like the Rectus femoris and Vastus lateralis linked to sarcopenia in men only. The study's small sample size calls for further research with more diverse and gender-balanced groups to verify these results.

## Introduction

Grip strength is more than just a measure of hand muscle capability; it serves as a vital indicator for overall physical performance and functional independence [1–3]. This aspect of human physiology becomes increasingly significant in day-to-day activities, facilitating takes ranging from carrying groceries to opening jars. The importance of grip strength is amplified in the elderly, where it often serves as a predictor for overall health outcomes [4].

Sarcopenia, the age-related loss of muscle mass and strength, presents a growing concern in the generalized loss of skeletal muscle mass and strength, significantly impacting the quality of life in older adults [5–7]. The consensus for diagnosing and assessing sarcopenia has been evolving, while traditionally focusing on the loss of skeletal muscle mass, recent trends in the field are moving towards including functional performance measures. In 2019, both the European Working Group on Sarcopenia in Older People 2 (EWGSOP2) and the Asian Working Group for Sarcopenia (AWGS2019) incorporated grip strength as a consensus measure for diagnosing and assessing the severity of sarcopenia [8,9].

Beyond the focus on the hand and forearm muscles in grip strength, other muscle groups like those in the thigh also play an essential role [10]. A loss of balance and stability could potentially compromise the ability to exert force effectively through one's grip [11]. The gluteus muscles are integral for maintaining balance [12]. Drawing on the insights from these studies, balance-critical muscles, including the lower body and thigh muscles, may have a secondary effect on grip strength by promoting enhanced stability and strength. Based on the findings from these studies, it is hypothesized that lower body muscles, including those in the thigh, contribute to stability and power, which could have an indirect effect on grip strength.

Acknowledging this complexity interplay between various thigh muscle groups and the evolving understanding of sarcopenia, the purpose of this study is to discover the relationship between individual thigh muscle and grip strength. This insight is crucial for understanding sarcopenia, where muscle mass and functional metrics such as grip strength play key roles. This emphasize on muscle mass and function highlights the necessity of precise individual muscle size measurement in a functional context.

While Cross-Sectional Area (CSA) in CT (Computed Tomography) scans has been traditionally used to assess muscle size, this method comes with limitations. Specifically, Cross-Sectional Area measurements, taken in a single plane, may not fully capture the intricacies of individual muscle groups, such as composition, distribution. Additionally, factors like participant positioning and choice of plane in image can affect the comparability of these measurements [13].

To overcome these shortcomings, we turn to muscle segmentation as a more comprehensive measure of individual thigh muscle volume. However, manual segmentation methos are both time-consuming and labor-intensive. To mitigate these challenges, we leverage AI-based automatic segmentation methods techniques, specifically employing UNETR, to assess individual thigh muscle volumes more efficiently and accurately [14].

The UNETR model represents an AI-driven semantic segmentation architecture that merges the U-net and Transformer modes. It excels in accurate voxel-level segmentation of 3D medical images, including CT scans and MRIs. In our previous study, we utilized the UNETR architecture to develop a thigh muscle segmentation model, employing a dataset of 72 entries (60 for training and 12 for validation) from the Real Hip Cohort. Real Hip Cohort includes variety of medical images, such as CT scans, X-rays and Dexa, along with physical performance metrics like grip strength from hip fracture patients.

The primary aim of this study is to investigate the relationship between individual thigh muscle volumes and grip strength in diagnosing sarcopenia. To achieve this, we employ a UNETR-based segmentation approach, enhancing the efficiency and precision in measuring the volumes of individual thigh muscles.

## Methods

### Participants

The study adhered to the principles of the Declaration of Helsinki and was approved by the IRB (IRB No. GNUH 2022-01-032-008) at Gyeongsang National University Hospital. All research procedures were carried out with strict adherence to ethical standards, including protection of participants' privacy, confidentiality, and rights. Written consent was obtained from the participants regarding their agreement to take part in the study.

In the cohort of hip fracture patients at Gyeongsang National University hospital, we screened 49 individuals from a pool of 478. These selected participants, who were part of the study from December 2016 to June 2022, had undergone CT scans and had their grip strength and height recorded. The research data were accessed on February 23, 2023 and the period of ethical approval for the study spanned from May 9, 2022 to May 8, 2023.

### Study design

In our study, we utilized a trained AI model to execute segmentation of individual thigh muscles, spanning from the hip to knee (whole thigh level), within CT scans. The goal was to investigate the relationship between individual thigh muscle volumes and grip strength. By employing the advanced deep learning framework UNETR, tailored for meticulous voxel-level segmentation and sequential data interpretation, we trained the model to achieve notable metrics with 72 CT scan datasets in our previous study. Specifically, our UNETR model attained a Dice score of 0.84 and a relative average volume difference of 0.019% [15]. These outcomes were corroborated by ground truth annotations from two medical radiologists, and the results of the segmentation process are presented in Fig 2.

One of the key analytical approaches we employ in this study is the point-biserial correlation method to examine the relationship between sarcopenia and individual thigh muscles. Importantly, we have adjusted the volume measurements of individual thigh muscles to account for variables like participant size and positioning, ensuring a more accurate and comparable analysis. Sarcopenia diagnosis in our study is based on grip strength (female<18kg and male<28kg) and skeletal muscle index using Dual-energy X-ray absorptiometry (female<5.4kg and male<7.0kg), following the guidelines set by Asian Working Group for Sarcopenia (AWGS), aligning with evolving consensus in the field that emphasizes functional performance measures alongside muscle mass [8]. By employing the point-biserial correlation method, we aim to provide a nuanced understanding of how individual thigh muscles, may be implicated in the onset or progression of sarcopenia when diagnosed through grip strength criteria.

Our study design also includes a comparative analysis of grouped thigh muscles and grip strength. For this purpose, participant's thigh muscles and grip strengths are categorized based on quantiles. This stratification enables us to explore the potential variations in grip strength across different quantile groups of thigh muscle volume. The central aim of this comparative analysis is to investigate the relationship between the balance of grouped thigh muscle volumes and the strength of grip. By examining this relationship, we seek to uncover patterns or trends that may offer valuable insights into how the proportional volumes of thigh muscles can influence grip strength. This is particularly relevant in the context of sarcopenia, where both muscle mass and functional performance measures like grip strength are of critical importance.

## CT scan acquisition

The present research included CT scans from 49 participants out of the cohort of 478 patients diagnosed with hip fractures. The average age of these participants was 77.3 years, with a standard deviation of 9.73 years. These individuals were enrolled from Gyeongsang National University Hospital during the period from December 2016 to June 2022.

As depicted in Fig 1 and Table 1, the inclusion criteria required that participants be in stable medical condition and have no contraindications against undergoing CT scans. In this study, the CT scans utilized were from participants underwent pre-operative CT scans, typically conducted within 24 hours following hip fracture incident. Additionally, as part of the inclusion criteria, both grip strength and height of the participants were measured.

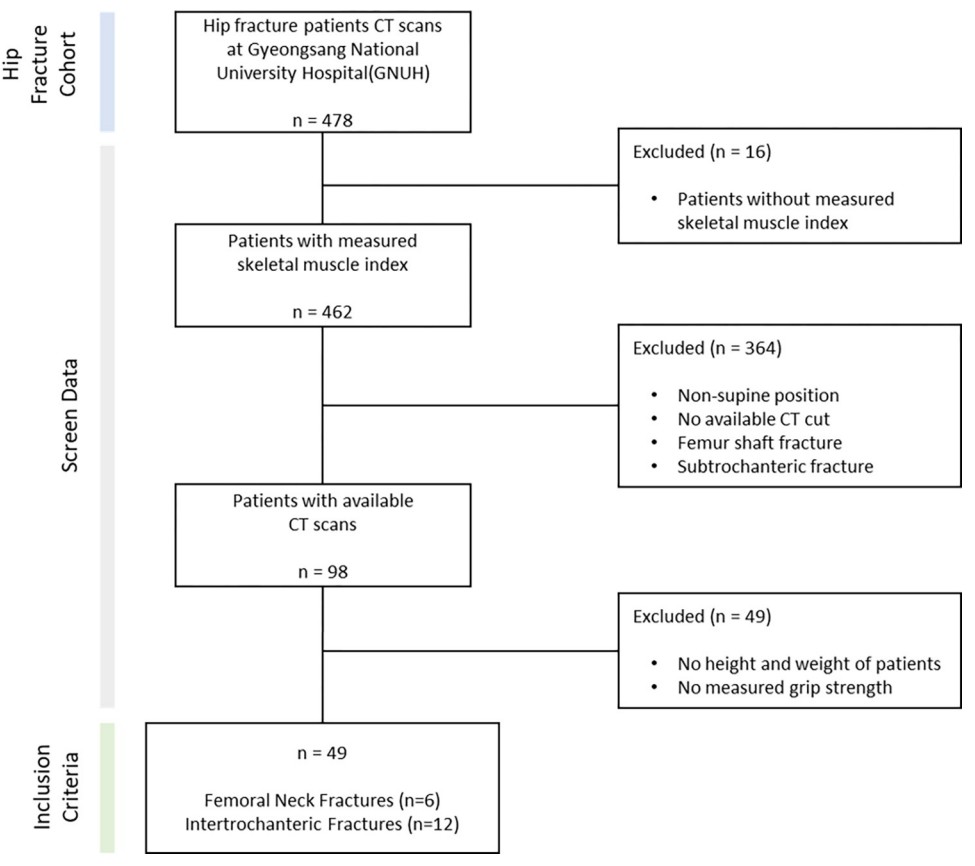

**Fig 1. Data screening.**

**Table 1. Inclusion and exclusion criteria.**

| Inclusion criteria | Exclusion criteria |
|---|---|
| • Hip fracture patients.<br>• Participants aged 65 years and above.<br>• Availability of pre-operative CT scans within 24 hours following hip fracture incident.<br>• Availability of weight and height.<br>• Availability of grip strength and skeletal muscle index measurements as consensus markers for sarcopenia | • CT scans taken in a non-supine position (folded legs or twisted body positions).<br>• Insufficient CT cuts to capture the entire thigh region (from hip to knee joint).<br>• Femur shaft fracture was observed.<br>• Observation of femur shaft fracture or subtrochanteric fracture.<br>• Presence of high artifacts or noise in CT scan images. |

This table outlines the criteria used for including and excluding participants in the study. Participants had to meet specific conditions, such as having undergone a hip fracture and pre-operative CT scans, as well as the availability of key clinical measurements like grip strength and skeletal muscle index. Exclusion criteria include factors that could impair the precision of muscle segmentation in CT scans, such as non-supine positioning or fractures in the femur shaft.

To uphold the precision and dependability of muscle segmentation on the CT scans, we set forth specific exclusion criteria. These encompassed conditions such as lower limb amputation, fractures in the shaft or subtrochanteric region of the femur, along with any discernible muscle or bone deformities, and significant metallic artifacts in the imaging.

Cases involving lower limb amputations were omitted as they could bring about considerable anatomical discrepancies that might compromise the accuracy of muscle segmentation. Similarly, individuals with femur shaft fracture or subtrochanteric fracture were excluded, as the angulation between the broken femur segments in CT scans could introduce muscle distortion. Metal artifacts in the imaging were also left out of the study to preserve the integrity and reliability of the muscle segmentation process in CT scans.

## Ground truth labeling

In the current research, we carried out a categorization that includes a total of 30 distinct classes. These encompass key elements such as femur, iliac and background classes that all derived from CT scans that cover the entire thigh region from the hip to the knee. We further segmented these scans into five primary thigh muscle areas: Anterior, Medial, Posterior and Gluteal region.

Within the anterior thigh region, we identified 5 specific muscles: Sartorius, Rectus femoris and Vastus group (lateralis, intermedius and medialis). In the medial section of the thigh, we distinguished 5 muscles, Adductor group (magnus, brevis and longus), Gracilis and Pectineus. The posterior thigh muscles were categorized into Semitendinosus, Semimembranosus and Biceps femoris. For the gluteal region, we classified 8 distinct muscles, including Gluteus group (maximus, medius and minimus), Fascia lata, Piriformis, Quadratus femoris, Obturator internus and Obturator externus.

Additionally, we made separate classifications for other muscles and elements within the image, such as the Iliacus, Iliopsoas, Psoas, Abdominal oblique, Rectus abdominis, Multifidus as well as the Femur and Iliac. A visual representation of these categorized elements in 3D is provided in Fig 2.

## CT scan image pre-processing

In the preliminary stage of data preparation, we employed a variety of heuristic strategies to boost the efficacy of deep learning algorithms in visual analyses. The initial step consisted of

| Anterior | Medial | Gluteal | Posterior | Else |
|----------|--------|---------|-----------|------|

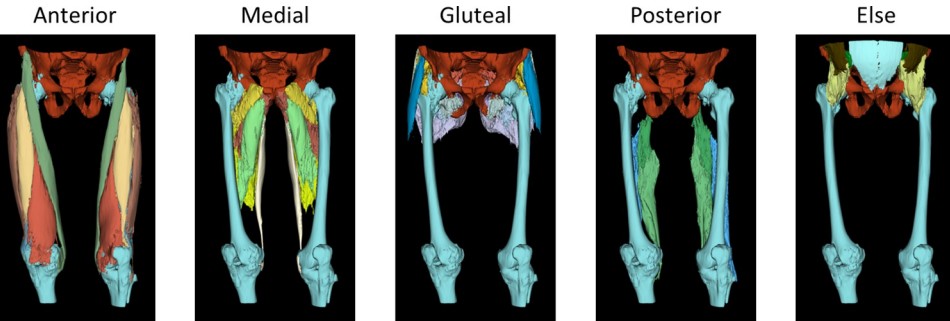

**Fig 2. Ground truth image in 3D.**

adjusting the CT scan image intensity levels from -57 to 164 to improve the discernibility of separate muscle tissues in CT scans [16,17]. Following this recalibration, we used a gamma correction with a value of 2 to further refine the image's contrast. The resultant image, notable for its enhanced contrast and minimized metallic distortions, is showcased in Fig 3. To zero in on pertinent details, we trimmed the image to include only the foreground.

## Methods

Our research concentrates on the crucial function of semantic segmentation, a fundamental aspect of computer vision that is particularly vital in the realm of medical imaging. The objective of this technique is to identify and classify separate areas of interest within an image. When applied to muscle segmentation, it requires labeling each voxel in the image as belonging to a specific class, such as muscle or bone tissue, or background. The challenge lies in capturing the intricate details and variations present in the images, like differing patient sizes, positions, and tissue characteristics, while also dealing with noise and other image imperfections. The often ambiguous nature of muscle tissue in CT scans makes this task even more demanding, requiring exact voxel-level segmentation. To tackle these complexities, we utilized the UNETR model framework, as illustrated in Fig 4.

The UNETR architecture leverages the power of transformers, which have shown exceptional performance in sequence-based domains, including Natural Language Processing (NLP). By framing 3D medical image segmentation as a sequence-based task, the UNETR model employs a transformer-based encoder to extract sequence representations from the

| | 160th slice | 120th slice | 80th slice | 60th slice |
|---|-------------|-------------|------------|------------|

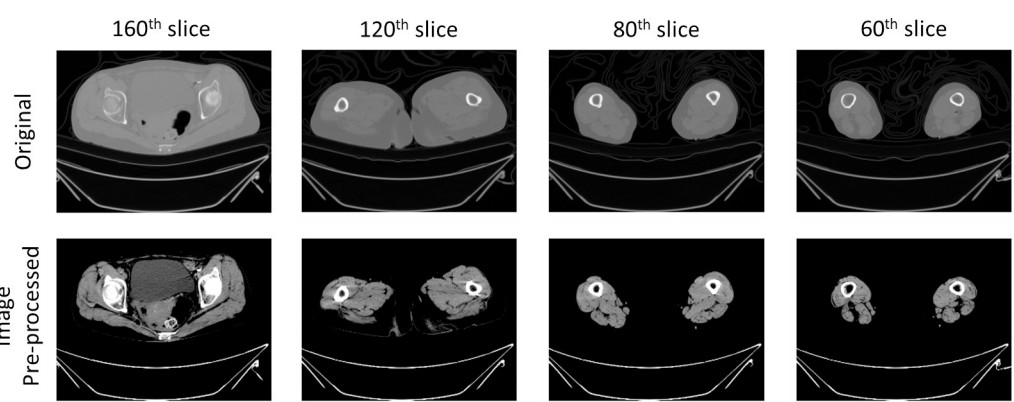

**Fig 3. Example of pre-processed CT scan image.**

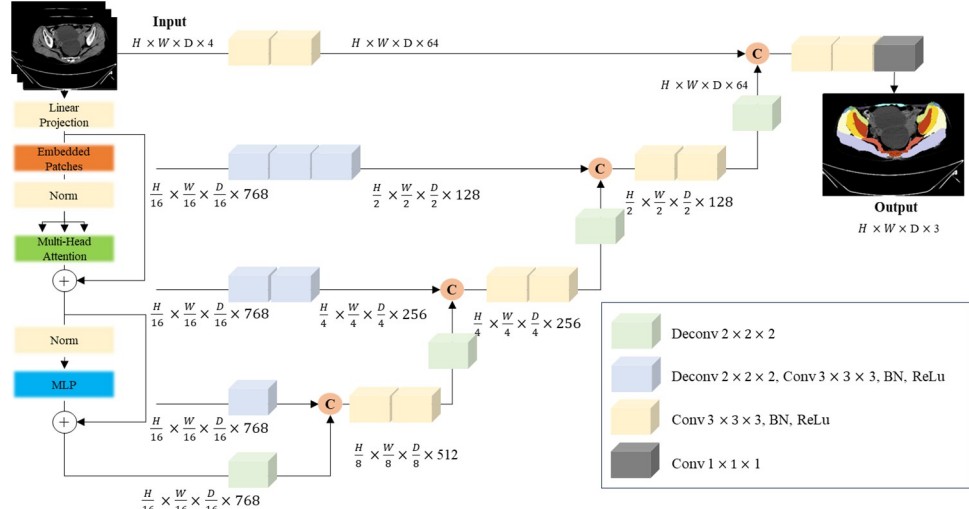

**Fig 4. UNETR model architecture.**

input volume, thereby capturing information across multiple scales. The encoder's structure takes inspiration from the U-shaped design of the original U-net model, well-known for its effectiveness in biomedical image segmentation. This design enables the model to capture both broad contextual information and fine-grained spatial details, making it highly suitable for segmenting individual thigh muscles.

Employing the cutting-edge UNETR architecture, specially crafted for precise voxel-level segmentation and sequence-based information processing, we trained our model to achieve a dice score of 0.81 and a relative average volume difference of 0.032±0.095. These results were validated against ground truth annotations from two radiologists, and the segmentation outcomes are displayed in Fig 5.

The muscle segmentation model was trained using an Nvidia DGX A100 system, powered by two Nivia A100 gpus and running on Ubuntu 20.04 operating system. The model was

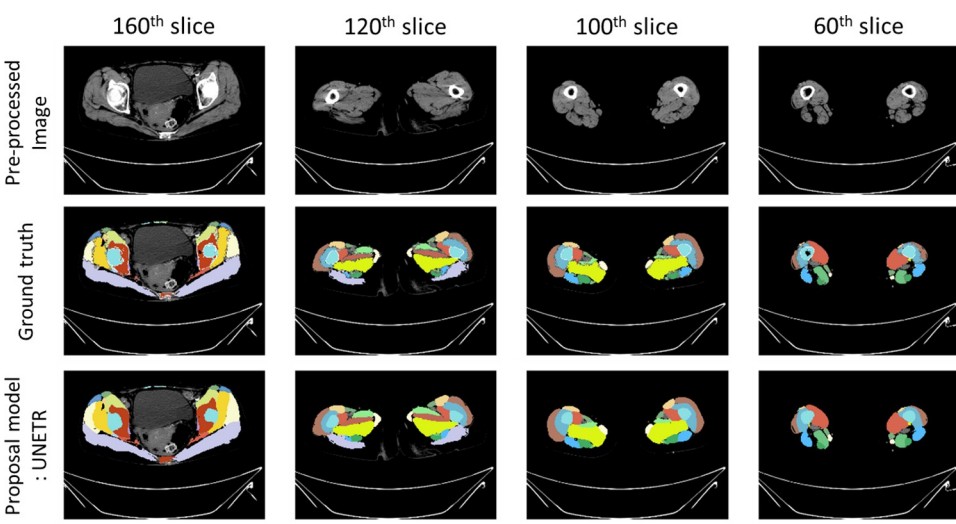

**Fig 5. Prediction example image of trained UNETR-based segmentation model.**

implemented with the Pytorch and MONAI libraries. For the training process, the dice coefficient was utilized as the loss function. The AdamW optimizer was set with a learning rate of 8e-5, weight decay of 1e-5, a batch size of 4 and 3000 epochs. These values were selected through a grid search method, testing different learning rates range from 1e-3 and to 1e-5.

The individual muscle volumes were derived from CT scans using trained UNETR model, to ensure linearity in the Skeletal Muscle Index (SMI) for each patient, we made further adjustments to the individual thigh muscle volumes. These adjustments were calculated by dividing the measured volume (measured in $mm^3$) by the $height^2$ (measured in mm).

Participants were grouped by gender and Point-biserial correlation was employed to calculate the correlation between sarcopenia and individual thigh muscle volumes. The point-biserial correlation coefficient is a statistical measure that quantifies the relationship between a continuous variable and a binary (dichotomous) variable. It is a specialized case of the Pearson correlation coefficient, used when one variable is continuous and normally distributed, and the other is dichotomous [18].

$$r_{pb} = \frac{M_1 - M_0}{s_n} \sqrt{\frac{n_1 n_0}{n^2}}$$

$$s_n = \sqrt{\frac{1}{n} \sum_{i=1}^{n} (x_i - \bar{x})^2}$$

- $M_1$ and $M_0$ are the means of the continuous variable for the two groups defined by the binary variable.

- $s$ is the standard deviation of the continuous variable for all participants.

- $n_1$ and $n_0$ are the number of cases in each of the two groups defined by the binary variable.

- $n$ is the total number of cases.

This statistical method allowed us to examine the relationship between grip strength and sarcopenia.

Furthermore, we conducted an analysis comparing the quantile distribution of grip strength in our datasets. This analysis was aimed at understanding how grip strength varies across different groups of thigh muscle volume, especially in the context of sarcopenia. For assessing balance, we combined muscle groups in specific ways, such as multiplying the anterior and medial groups, as well as the posterior and gluteal regions. We also examined the similarities between these combined muscle groupings.

$$similarity = \frac{1}{1 + |a - b|}$$

## Results

In this study, the participant cohort consisted of individuals with hip fractures, totaling 49 patients. Among them, 32 were females, while 17 were males. The average age of the female group was 77.3 years, with a standard deviation of 10.46, while the average age of the male group was 75.6 years, with a standard deviation of 6.42.

In the current investigation, utilizing grip strength (with under threshold set at 18kg for females and 28kg for males) and skeletal muscle index (under 5.4kg for females and 7.0kg for males) as a consensus measure for diagnosing sarcopenia, we identified 43 patients with

**Table 2. Point-biserial correlation between sarcopenia diagnosed with grip strength and each individual thigh muscle volume.**

| Muscle | Female | | Male | |
|---|---|---|---|---|
| | Correlation coefficient | p-value (*p<0.05) | Correlation coefficient | p-value (*p<0.05) |
| Sartorius | 0.173 | 0.342 | -0.479 | 0.051 |
| Rectus Femoris | 0.007 | 0.969 | -0.614 | < 0.001 ** |
| Vastus Lateralis | -0.020 | 0.913 | -0.694 | < 0.001 ** |
| Vastus Intermedius | 0.080 | 0.664 | -0.671 | < 0.001 ** |
| Vastus Medialis | 0.013 | 0.942 | -0.560 | < 0.05 * |
| Adductor Longus | 0.067 | 0.714 | -0.526 | < 0.05 * |
| Adductor Brevis | 0.276 | 0.127 | -0.486 | < 0.05 * |
| Adductor Magnus | 0.034 | 0.852 | -0.524 | < 0.05 * |
| Gracilis | 0.138 | 0.451 | -0.429 | 0.086 |
| Pectineus | 0.254 | 0.161 | -0.526 | < 0.05 * |
| Gluteus Maximus | 0.128 | 0.483 | -0.521 | < 0.05 * |
| Gluteus Medius | 0.039 | 0.834 | -0.231 | 0.372 |
| Gluteus Minimus | 0.063 | 0.734 | -0.369 | 0.145 |
| Tensor Fascia Lata | 0.164 | 0.369 | -0.544 | < 0.05 * |
| Piriformis | 0.236 | 0.194 | -0.342 | 0.179 |
| Obturator Internus | 0.016 | 0.93 | -0.445 | 0.074 |
| Obturator Externus | 0.021 | 0.907 | -0.361 | 0.155 |
| Quadratus Femoris | 0.253 | 0.163 | -0.337 | 0.186 |
| Semitendinosus | 0.094 | 0.61 | -0.543 | < 0.05 * |
| Semimembranosus | 0.020 | 0.914 | -0.212 | 0.415 |
| Biceps Femoris | 0.064 | 0.726 | -0.378 | 0.134 |

*: p < 0.05

**: p < 0.01

***: p < 0.001.

This table presents the point-biserial correlation coefficients and p-values assessing the relationship between sarcopenia (diagnosed via grip strength) and the volume of individual thigh muscles. Significant correlations were predominantly observed in the male cohort, with certain muscles such as Rectus femoris, Vastus lateralis, and Vastus intermedius displaying strong associations. No significant correlations were found for the female cohort.

sarcopenia. This group comprised 28 females and 15 males, while 6 patients were not diagnosed with sarcopenia, including 4 females and 2 males.

Upon adjusting for individual thigh muscle volumes, our initial step was to employ the point-biserial correlation method to explore relationship between sarcopenia and specific thigh muscle volumes. As presented in Table 2, significant findings were predominantly observed in the male cohort. Specifically, the Rectus femoris, Vastus lateralis and Vastus intermedius displayed strong statistical significance (p < 0.001**). Additionally, Vastus medialis, Adductor group (longus, brevis and magnus), Pectineus, Gluteus maximus, Tensor fascia lata and Semitendinosus were also statistically significant (p < 0.05*). Conversely, no significant correlations were observed between sarcopenia and any individual thigh muscle in the female group.

Additionally, we conducted Spearman's rank correlation analysis to determine the relationship between grip strength and the volume of each thigh muscle individually. The results, presented in Table 3, show a significant correlation between grip strength and the volumes of the Rectus femoris, Gluteus maximus, and Semimembranosus (all with p < 0.01**) in the male

**Table 3. Spearman's rank correlation between grip strength and each individual thigh muscle volume.**

| Muscle | Female | | Male | |
|---|---|---|---|---|
| | Spearman R | P-value | Spearman R | P-value |
| Sartorius | 0.124 | 0.499 | 0.554 | < 0.05 * |
| Rectus Femoris | 0.188 | 0.303 | 0.63 | < 0.01 ** |
| Vastus Lateralis | 0.234 | 0.198 | 0.571 | < 0.05 * |
| Vastus Intermedius | 0.078 | 0.67 | 0.505 | < 0.05 * |
| Vastus Medialis | 0.247 | 0.173 | 0.397 | 0.115 |
| Adductor Longus | 0.101 | 0.584 | 0.412 | 0.101 |
| Adductor Brevis | 0.098 | 0.594 | 0.532 | < 0.05 * |
| Adductor Magnus | 0.161 | 0.379 | 0.478 | 0.052 |
| Gracilis | -0.033 | 0.859 | 0.522 | < 0.05 * |
| Pectineus | 0.036 | 0.843 | 0.515 | < 0.05 * |
| Gluteus Maximus | 0.105 | 0.567 | 0.647 | < 0.01 ** |
| Gluteus Medius | 0.083 | 0.653 | 0.172 | 0.51 |
| Gluteus Minimus | 0.052 | 0.779 | 0.24 | 0.353 |
| Tensor Fascia Lata | 0.171 | 0.35 | 0.35 | 0.168 |
| Piriformis | 0.079 | 0.666 | 0.466 | 0.06 |
| Obturator Internus | 0.135 | 0.46 | 0.4 | 0.112 |
| Obturator Externus | 0.057 | 0.756 | 0.439 | 0.078 |
| Quadratus Femoris | -0.134 | 0.466 | 0.468 | 0.058 |
| Semitendinosus | 0.054 | 0.769 | 0.473 | 0.055 |
| Semimembranosus | 0.248 | 0.171 | 0.689 | < 0.01 ** |
| Biceps Femoris | 0.164 | 0.369 | 0.424 | 0.09 |

*: $p < 0.05$

**: $p < 0.01$

***: $p < 0.001$.

This table shows the results of Spearman's rank correlation analysis between grip strength and the volume of individual thigh muscles for both male and female cohorts. Significant positive correlations were noted in males for several muscles, including Rectus femoris, Gluteus maximus, and Semimembranosus. No significant correlations were observed for the female group.

cohort. Furthermore, significant correlations were observed for the Sartorius, Vastus lateralis, Vastus intermedius, Adductor brevis, Gracilis, and Pectineus muscles (each with $p < 0.05$*) within this group. Conversely, no significant correlation was found between grip strength and the volume of any individual muscle in the female cohort.

In our quantile analysis of grip strength, the male cohort displayed a mean grip strength of 21kg with a standard deviation of 7.5kg. The first quantile (Q1) was at 18kg, the median (Q2) stood at 20kg, the third quantile (Q3) was at 26.3kg and the fourth quantile (Q4) was 37kg. In contrast, the female cohort had a mean grip strength of 12kg, accompanied by a standard deviation of 5.14kg. The first quantile (Q1) was at 5kg, the median (Q2) was at 11.6kg, the third quantile (Q3) was 14.72kg and the fourth quantile (Q4) was 27.6kg.

Subsequent to adjusting for muscle volumes by height (volume/height$^2$), as illustrated in Fig 6, a trend was observed exclusively within the male group, where greater muscle volumes were associated with higher grip strengths. Furthermore, as shown in Fig 7, the anterior muscle group displayed specific mean volumes across different quartiles: 68.95 in the second quartile (Q2), 93.83 in the third quartile (Q3), and 107.02 in the fourth quartile (Q4). These findings suggest a subtle yet discernible trend between anterior muscle volume and grip

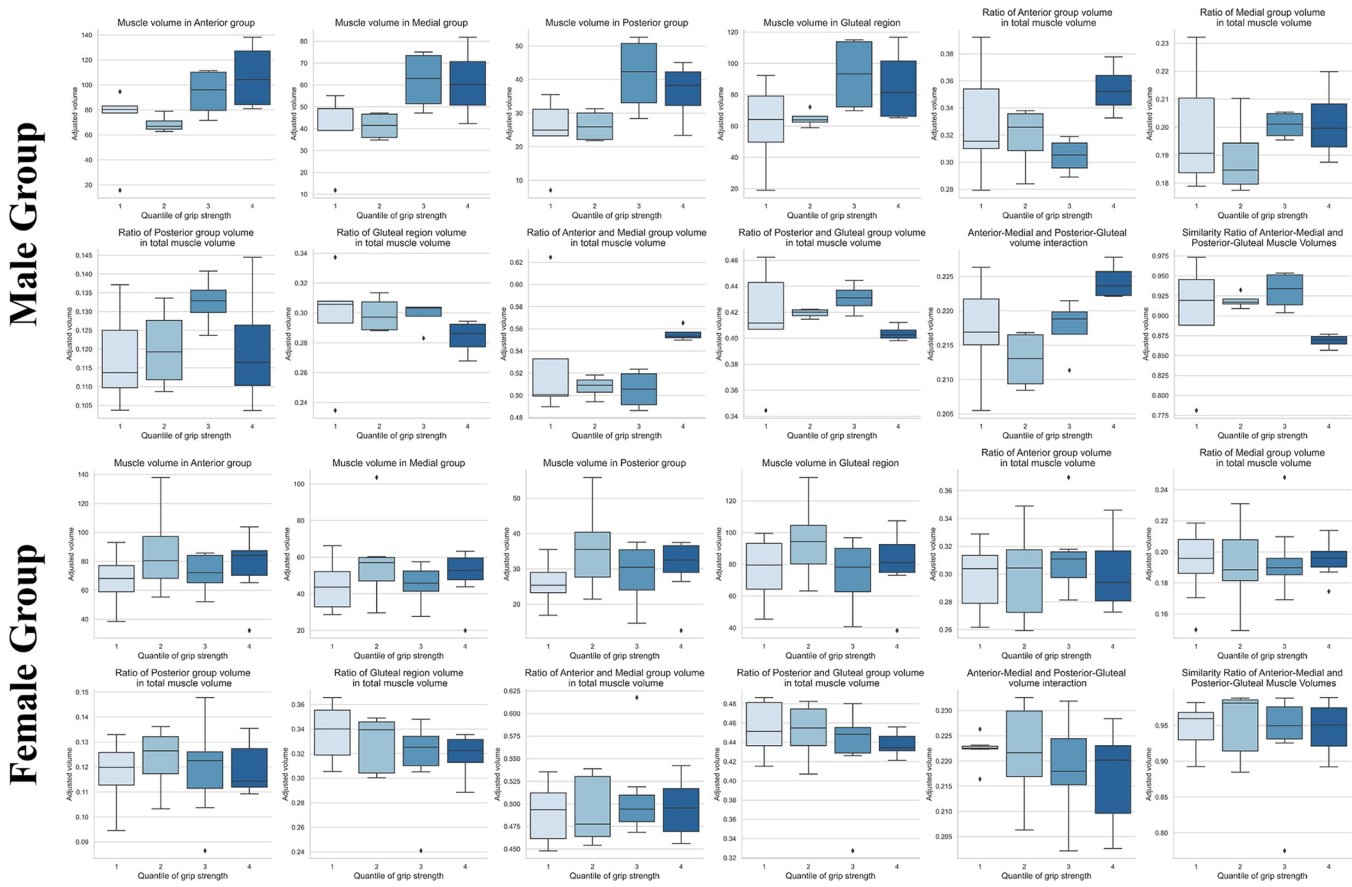

**Fig 6. Box plots of grouped thigh muscles and quantile grip strength.**

strength in male group. This pattern was not observed in the female group, where no significant relationship with muscle volume was found (p > 0.05).

Moreover, as illustrated in Fig 6, there is a notable difference between the third (Q3) and fourth quantiles (Q4) in the ratio of (anterior + medial) × (posterior + gluteal region). Additionally, the similarity between (anterior + medial) and (posterior + gluteal region) appears to be distinct. These observations suggest, especially within older male demographic, maintaining an appropriate anterior muscle volume ratio has trend of stronger grip strength.

# Conclusion

## Discussion

Grip strength serves as a vital physiological marker and is often considered an indicator of overall muscle health. Its importance is particularly emphasized in the context of aging populations where reduced grip strength can be a red flag for various health-related issues, including sarcopenia [19–22].

The present study employed grip strength and skeletal muscle index as a consensus marker for diagnosing sarcopenia and found significant correlations between sarcopenia and thigh muscle volumes such as Rectus femoris, Vastus lateralis and Vastus intermedius, particularly in the male cohort. Those specific thigh muscles imply their potential contribution in the onset or progression of sarcopenia in males. Conversely, no significant associations were observed between sarcopenia and each individual muscle volume in female group.

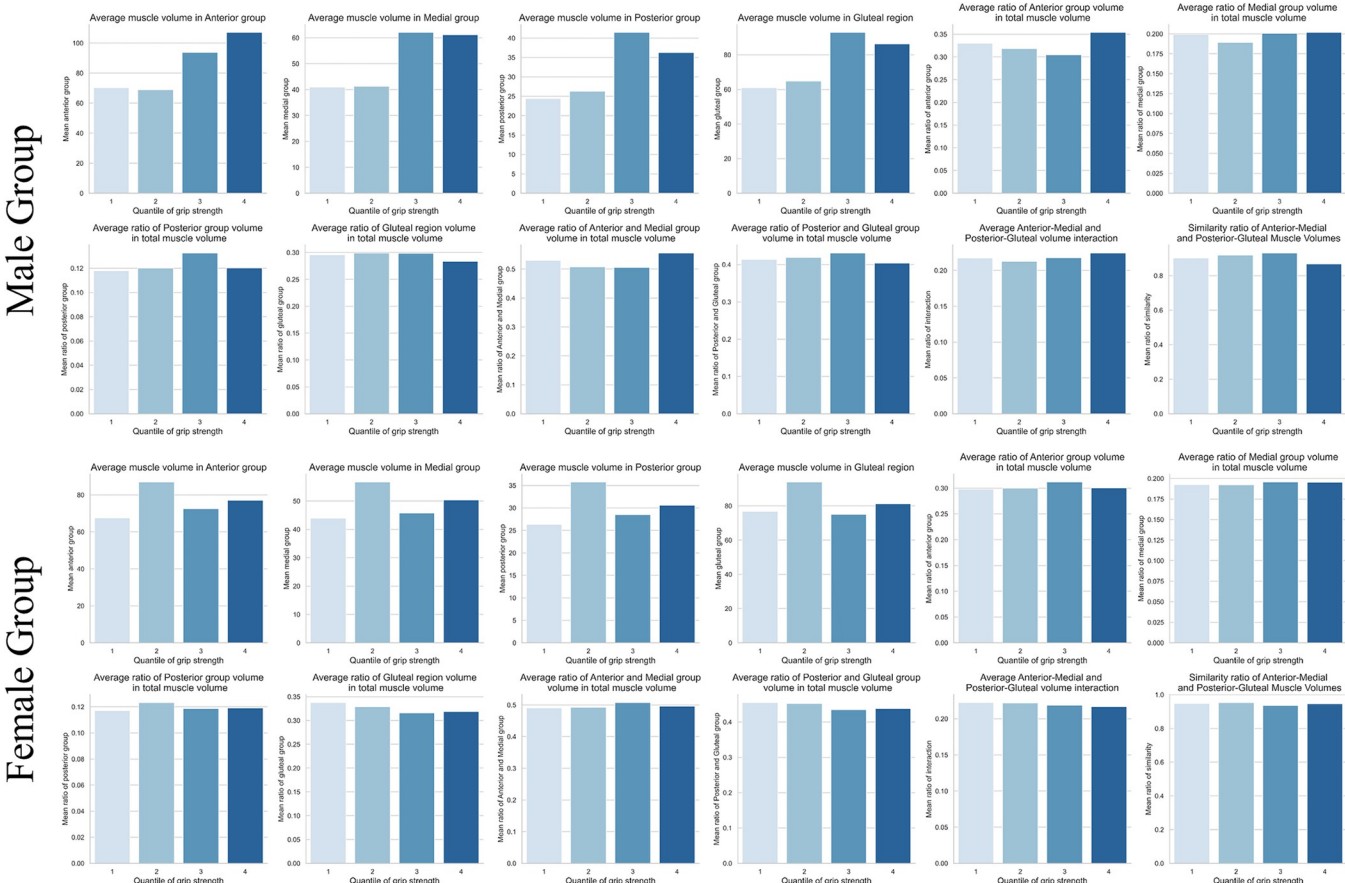

**Fig 7. Grouped thigh muscle distribution in quantile grip strength.**

The Spearman's rank correlation analysis revealed significant correlations between grip strength and the volumes of several thigh muscles in the male group, specifically the Rectus femoris, Gluteus maximus, and Semimembranosus (all $p < 0.01^{**}$), as well as the Sartorius, Vastus lateralis, Vastus intermedius, Adductor brevis, Gracilis, and Pectineus muscles (each with $p < 0.05^*$). However, in the female group, there was no significant correlation between grip strength and the volume of any individual muscle. This differentiation in correlation patterns between genders could offer insightful perspectives for further discussions on physiological differences and their implications.

The analysis of quantile indicates a trend within the male group where increased volumes of the anterior and medial thigh could be linked with stronger grip strength. Intriguingly, the study also indicated that maintaining an appropriate ratio of anterior to medial and posterior to gluteal muscle volumes could be particularly beneficial for grip strength, especially among older males.

The result indicates that the posterior thigh muscles may be underutilized in the everyday activities of older adults, hinting at potential muscle imbalances. This underscores the relevance of achieving a balanced muscle composition, particularly among the elderly. The findings also suggest the existence of a favorable ratio among grouped thigh muscles that could optimize muscle function and grip strength. Specifically, tailored exercises that target particular thigh muscle groups might offer more substantial benefits for maintaining stable gait and overall physical agility in older populations [23–25]. By encouraging such targeted physical

activity, we could potentially reduce the severity of sarcopenia, thereby improving life quality for aging individuals.

In the result of correlation between thigh muscle volume and sarcopenia, the lack of a significant in female group may stem from the interplay of hormonal mechanisms across genders. Regarding hip fractures, the timing of peak bone and muscle mass loss, leading to increased fragility of hip fractures after this peak intersection point [26–29]. For men, the sequence typically involves a decrease in muscle mass due to lower testosterone levels before bone loss occurs [30]. In contrast, in women, the process of muscle loss generally occurs after the loss of bone density, which is influenced by estrogen levels [31,32]. This suggests that for women, muscle mass might not significantly influence the progression of sarcopenia in the context of hip fractures. However, this conclusion could be affected by a potential bias due to a limited sample size in the study group.

## Limitation

Our study, while offering valuable insights into the relationship between thigh muscle volumes and grip strength, has several limitations. First, the sample size was relatively small and our findings were predominantly observed in males, which limit the generalizability results.

Secondly, the study primarily focused on patients with hip fractures, a group inherently linked to sarcopenia, hence lacking a comparative control group without hip fractures to fully evaluate these associations. This limitation, along with an imbalance in our dataset concerning sarcopenia diagnosis because hip fractures are more common in older individuals who are also more likely to have sarcopenia, could potentially lead to biased results [33–35].

Further research is warranted to determine whether the observed trends would persist in individuals without hip fractures, as such findings could potentially enhance the utility of these measures in diagnosing sarcopenia. In future studies, it would be useful to examine if the observed trends continue in individuals without hip fractures, as this could help refine the application of these metrics in diagnosing sarcopenia.

## Conclusion

The findings suggest a tendency towards stronger grip strength being associated with larger anterior and medial thigh muscles in the male group, whereas no significant trend was observed in the female group. Specifically, in males, certain thigh muscles such as the Rectus femoris, Vastus lateralis, Vastus intermedius, Adductor group, and Gluteus maximus correlated with sarcopenia, while females did not exhibit reasonable correlations. While grip strength may have limited utility as a screening tool for sarcopenia in females, the small sample size in this study restricts the generalizability of these findings. Future research should aim to validate these observations in larger and more diverse cohorts with balanced gender representation to enhance the interpretability and applicability of the results.

## Author Contributions

**Conceptualization:** Hyeon Su Kim, Yonghan Cha, Jung-Taek Kim, Jin-Woo Kim, Yong-Chan Ha, Jun-Il Yoo.

**Data curation:** Hyeon Su Kim, Shinjune Kim, Hyunbin Kim, Jung-Taek Kim.

**Formal analysis:** Hyeon Su Kim, Shinjune Kim.

**Investigation:** Hyeon Su Kim, Jun-Il Yoo.

**Project administration:** Jun-Il Yoo.

**Software:** Hyeon Su Kim.

**Supervision:** Jun-Il Yoo.

**Validation:** Hyeon Su Kim, Shinjune Kim.

**Visualization:** Hyeon Su Kim.

**Writing – original draft:** Hyeon Su Kim.

**Writing – review & editing:** Shinjune Kim, Hyunbin Kim, Yonghan Cha, Jung-Taek Kim, Jin-Woo Kim, Yong-Chan Ha, Jun-Il Yoo.

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
