## [Decision Letter · Decision Letter 0]

16 Feb 2024

PONE-D-23-28695Correlation Between Individual Thigh Muscle Volume and Grip Strength

In Relation to Sarcopenia: An AI-based Muscle Segmentation Using UNETRPLOS ONE

Dear Dr. Yoo,

Thank you for submitting your manuscript to PLOS ONE. After careful consideration, we feel that it has merit but does not fully meet PLOS ONE’s publication criteria as it currently stands. Therefore, we invite you to submit a revised version of the manuscript that addresses the points raised during the review process. Both Expert Reviewers raised a number of concerns ranging from technical issues to better explanations of the approaches for analyses and some of the comparisons for example between thigh mass and forearm strength. If you go carefully through each of the points raised by the Reviewers you will see ways that your manuscript can be strengthened and improved.

We look forward to receiving your revised manuscript.

Kind regards,

Stephen E Alway, Ph.D.

Academic Editor

PLOS ONE

Journal Requirements:

6. Please include your tables as part of your main manuscript and remove the individual files. Please note that supplementary tables (should remain/ be uploaded) as separate "supporting information" files

Reviewers' comments:

Reviewer's Responses to Questions

**Comments to the Author**

1. Is the manuscript technically sound, and do the data support the conclusions?

Reviewer #1: Partly

Reviewer #2: Yes

2. Has the statistical analysis been performed appropriately and rigorously? 

Reviewer #1: No

Reviewer #2: Yes

3. Have the authors made all data underlying the findings in their manuscript fully available?

Reviewer #1: Yes

Reviewer #2: Yes

4. Is the manuscript presented in an intelligible fashion and written in standard English?

Reviewer #1: Yes

Reviewer #2: Yes

5. Review Comments to the Author

Reviewer #1: The article “Correlation Between Individual Thigh Muscle Volume and Grip Strength

In Relation to Sarcopenia: An AI-based Muscle Segmentation Using UNETR” adds the evidence of using AI-based segmentation of CT for value-added screenings. The authors show value by pushing limits of using AI in segmentation, especially one based on a transformer model, in clinical cases. This will help push the field and give an example of a segmentation algorithm that can assess and label multiple muscle groups. However, this reviewer has found multiple concerns that should be addressed before publication.

Minor concerns

• What are limitations of focusing only on grip strength as a functional outcome? EWGSOP does also mention other functional measures. Thigh volume correlation to grip strength is not as important, at least to this reviewer, as the relationship between thigh volume correlation and sarcopenia. Consider this as you write both the introduction and discussion.

• What about CT muscle density? Segmentation models exist to automatically evaluate muscle density, i.e. muscle quality, which has a higher association with sarcopenia than volume, i.e. muscle mass. This should be addressed as a potential limitation of the study as recommendations by EWGSOP prefer muscle quality over muscle mass.

• Change section title “Result” to “Results”

• Change “groped” to “grouped” for Fig 7 title

• The resolution of Fig 6 and 7 are low. A lot of information is packed into these figures, so when I zoom in to see trends and read labels, the font becomes blurry.

• Technical features of segmentation model. Did any cases fail or need to be run again? What is the run time for the analysis of your images? This can help advocate for the implementation of your model in a practical setting with real-world images.

• The discussion does not connect much to established literature.

Major concerns

• AI is not the main focus of this article or being tested. Consider changing the title as it makes the article seem more about using AI in a novel way instead of an existing segmentation model.

• Patients all have hip fractures, which is associated with sarcopenia. There may not be a true control group to test these associations fully. Would the trends still exist with patients without hip fractures? If so, these associations might be fully helpful in diagnosing sarcopenia.

• Not clear in the first paragraph of the study design if the UNETR model was previously trained in the cited article or trained using the current sample.

• In the results of the point biserial correlation, the p-values are important but do not show the strength of correlation. What is the correlation value between the muscle groups and sarcopenia? Please report the correlation values in addition to the p-values. Without these, I cannot assess the strength of relationships between sarcopenia and thigh muscles.

• There are no y-axis labels in Fig 6.

• In Fig 6 and Fig 7, change your titles and x-axis to human readable labels instead of variable names.

• Results section for quantiles of men and women define 3 quartiles instead of 4 as displayed in Fig 6 and 7.

• Fig 6 and 7 are unclear if adjusted or raw muscle volumes are being used for correlations. State this is in the results as well.

• In results, please note that the noteworthy trend was not test statistically.

• In results, note the lack of trends among women’s muscle volume and grip strength. This is important finding especially as EWGSOP is recommending muscle performance for sarcopenia diagnosis.

• In results, the sentence “These findings suggest a subtle yet discernible correlation between anterior muscle volume and grip strength.” Should be changed to “trend between…” as no correlation was tested.

• A lot of the issues of “trends” in Figure 6 and 7 could be tested by comparing the muscle group volumes and ratios to grip strength continuously. You would be able to assess trend strength instead observing possible trends. You could also represent these in tables, which would likely make your results more readable.

• The last paragraph of the results states there are differences and associations, but none have been tested statistically between grip strength and muscle volumes. This reviewer highly recommends changing this language to “trends” or testing them with Pearson’s correlations and/or ANOVA tests.

• Please clarify the second statement in the discussion. Is this a correlation between muscle volumes and grip strength or muscle volumes and sarcopenia? If this is reference to muscle volumes and grip strength, that association was not tested as far as this reviewer can tell.

• The statement in the discussion- “The quantile analysis further supported these findings showing a clear trend in the male group that higher muscle volumes are correlated with better grip strengths.”- is not supported in the research. No correlation was established statistically.

• This reviewer would also like to see the point biserial correlations and any of the muscle volume comparisons of the ground truth against the AI segmentation to see how the differences compare to a performance metric (grip strength) and diagnosis of sarcopenia. If this is included, this reviewer feels the title including AI would be more valid.

Reviewer #2: This article aimed to investigate the correlation between individual thigh muscle volumes and grip strength, utilizing advanced AI-based UNETR segmentation techniques for accurate assessment of muscle volume. The study specifically focused on a sample of hip fracture patients, utilizing point-biserial correlation to explore the relationship between sarcopenia and thigh muscle volumes. Overall, the study successfully introduced the concept of examining the balance of grouped thigh muscle volumes and its association with grip strength by incorporating cutting-edge AI technology. The findings revealed a positive relationship between thigh muscle volumes and grip strength. However, several limitations should be acknowledged, including the relatively small sample size and the imbalance in the dataset regarding sarcopenia diagnosis. These limitations may impact the validity of the findings. The study addresses a research gap in the literature and is novel in using the UNETR technique. However, major concerns regarding the rationale and methodology are noted and identified below.

Introduction:

In the introduction, it would be beneficial to include a brief explanation of UNETR and why it is a more accurate choice for assessing individual thigh muscle volumes. This will help to avoid any potential confusion.

The introduction lacks an explanation and reference for why the thigh muscles have a direct impact on grip strength (Ln 65-67). This oversight makes it difficult to understand the primary aim mentioned later (Ln 83-85), particularly the relationship between thigh muscle volumes, grip strength, and sarcopenia, as well as the reason for investigating these factors.

Ln 116-118, “we aim to provide a nuanced understanding of how individual thigh muscles, once adjusted for volume, may be implicated in the onset or progression of sarcopenia when diagnosed through grip strength criteria.” Including this statement in the introduction would provide the audience with a more comprehensive understanding of the study's intentions.

Ln 124-127, “By examining this relationship, we seek to uncover patterns or trends that may offer valuable insights into how the proportional volumes of thigh muscles can influence grip 126 strength. This is particularly relevant in the context of sarcopenia, where both muscle mass and functional performance measures like grip strength are of critical importance” should also be mentioned concisely in introduction.

In Line 54-57, the two sentences repeat the same message and are nearly identical in terms of writing. The authors should consider amending the second sentence to provide reasons for why sarcopenia can significantly impact the quality of life in older adults.

In Line 62, the authors state that "EWGSOP2 and AWGS2019 incorporated grip strength as a consensus measure for diagnosing and assessing the severity of sarcopenia". This statement is inaccurate and is a critical concern of this manuscript. A key assumption of this study is that the association between thigh muscle and grip strength can inform the role of thigh muscle in sarcopenia due to grip strength being a consensus measure of sarcopenia. However, grip strength alone cannot be used to diagnose or assess the severity of sarcopenia. Indeed, the operational definition of sarcopenia according to the European Working Group on Sarcopenia (EWGSOP2) and the Asian Working Group for Sarcopenia (AWGS2019) requires low grip strength to be accompanied by either low muscle quantity or quality for EWGSOP2 and low appendicular skeletal muscle mass for AWGS2019. Please provide a detailed justification for why grip strength independently was used to diagnose sarcopenia in this study.

In Line 67-68, the authors state that "gluteus muscles are integral for maintaining balance, a factor that might also have an impact on grip strength". In fact, the referenced study does not include grip strength as an outcome nor mention grip strength in their article. The authors should justify this statement and amend the reference.

The authors have highlighted the limitations of previous methods to assess muscle size and promoted UNETR as a potential alternative. However, the authors did not explain what UNETR is, and why it might serve as a more valid and efficient method to assess muscle volume. Therefore, the rationale for using the UNETR technique is missing. As the research gap identified is to investigate the correlation of thigh muscle volume assessed via the UNETR technique with grip strength, it is crucial to identify the importance of using the UNETR technique.

Methods:

To ensure a comprehensive and unbiased account of the study's outcomes, it is essential to include and describe the findings of both the male and female cohorts in the results section. Even if the findings for the female cohort are nonsignificant, their inclusion provides a complete picture of the study's results and their potential implications for both genders.

In Line 94, the authors state that participants were from a "cohort of hip fracture patients". This is another critical concern, as hip fracture patients are likely to have experienced thigh muscle atrophy during their recovery period, and therefore have lower thigh muscle volume as a result of their injury. This can strongly bias the findings of the study. Please clarify and justify why participants were selected from a cohort of hip fracture patients.

In Line 113, the authors state that the diagnosis of sarcopenia is based on grip strength. As mentioned previously, the diagnosis of sarcopenia according to the AWGS2019 definition requires that the individual have low appendicular skeletal muscle mass coupled with low muscle strength (i.e., low grip strength). The authors should consider reassessing the presence of sarcopenia in their study sample while complying fully with the AWGS2019 definition.

In Line 134, the authors refer to Figure 1 for the inclusion criteria. Figure 1 is the flowchart of the study. The authors should include a table for eligibility, detailing the inclusion and exclusion criteria.

Results:

The authors should include the baseline demographic and characteristics of the participants.

Results for the correlation between sarcopenia (diagnosed using grip strength independently) and individual thigh muscles for the female group were not reported. This omission is misleading and may cause the reader to presume that thigh muscle volume is strongly correlated with sarcopenia when this finding only holds true in the male group. Indeed, upon reviewing Table 1, none of the individual thigh muscles was significantly correlated with sarcopenia in the female group. Please include the findings for the female group in the results section.

Discussion:

The discussion section recommends targeted physical activity to maintain stable gait and overall physical agility in older populations. However, the study did not assess gait or physical agility, and therefore the recommendation lacks support. The discussion would benefit from including a reference to support the relationship between thigh muscle composition and gait/physical agility to strengthen the recommendation.

The discussion section should further analyze and interpret the possible reasons for the non-significant result of the female cohort, such as sample size, hormonal differences, or other variables. This will provide valuable insights and contribute to the overall interpretation of the study's findings.

Due to the small sample size and the predominance of findings in males, the study's generalizability is limited. Describing the relationships as "promising" may be misleading. It is important to consider these limitations and interpret the findings cautiously.

In the "Discussion" section, the authors did not provide any explanation for why no correlation was observed between individual thigh muscles and sarcopenia diagnosis (based on grip strength alone) for the female group. The authors should discuss the potential reasons for why a significant correlation was observed in the male group but not the female group, and the potential implications of this finding.

6. PLOS authors have the option to publish the peer review history of their article (what does this mean?). If published, this will include your full peer review and any attached files.

Reviewer #1: **Yes: **Benjamin Rush

Reviewer #2: No

---

## [Author Response · Author response to Decision Letter 0]

11 Jun 2024

Response: 

Minor concerns

1. What are limitations of focusing only on grip strength as a functional outcome? EWGSOP does also mention other functional measures. Thigh volume correlation to grip strength is not as important, at least to this reviewer, as the relationship between thigh volume correlation and sarcopenia. Consider this as you write both the introduction and discussion.

Response: 

The consensus of EWGSOP diagnoses and determines the severity of sarcopenia by measuring grip strength, chair stand test, SPPB and gait speed to assess physical performance. The current guideline or diagnosing sarcopenia involves initially measuring grip strength and conducting a chair stand test to identify a ‘sarcopenia probable’ status, which is then confirmed through skeletal muscle index from Dual-Energy X-ray Absorptiometry (DXA). However, in our current study, we only had access to grip strength measurements in physical performance measurements for hip fracture patients. The majority of these patients, upon hospital admission, were unable to walk or undergo tests to evaluate their physical performance (SPPB, chair stand test and gait speed). So, we utilized grip strength to determine a ‘sarcopenia probable’ status and employed DXA scanning for the definitive diagnosis of sarcopenia. 

2. What about CT muscle density? Segmentation models exist to automatically evaluate muscle density, i.e. muscle quality, which has a higher association with sarcopenia than volume, i.e. muscle mass. This should be addressed as a potential limitation of the study as recommendations by EWGSOP prefer muscle quality over muscle mass.

Response:

We recognize the importance of assessing adipose tissue, both intermuscular and intramuscular, in the elderly and its impact on their physical function. However, our approach to developing the model involved using the labeled dataset, created by a threshold for fat tissues as background class. This process leads to a model that identifies muscle tissue exclusively, neglecting any fat present within the muscle. This limitation leads us to challenges to precisely evaluate the percentage of muscle fat infiltration in the outputs from our segmentation model. Currently, our research is focused on the volume of anatomical individual muscles at each functional site.

3. Change section title “Result” to “Results” 

Response: 

I appreciate your pin out. I have changed the section title of “Result” to “Results”

4. Change “groped” to “grouped” for Fig 7 title

Response:

Thank you for the pointing out the typo. I modified the title of Figure 7 to “Grouped thigh muscle distribution in quantile grip strength”

5. The resolution of Fig 6 and 7 are low. A lot of information is packed into these figures, so when I zoom in to see trends and read labels, the font becomes blurry.

Response: 

Thank you for your feedback. I have noticed that Figure 6 and 7 become blurry when I zoom in. I have uploaded the Figure 6 and 7 with higher resolution.

6. Technical features of segmentation model. Did any cases fail or need to be run again? What is the run time for the analysis of your images? This can help advocate for the implementation of your model in a practical setting with real-world images.

Response: 

I appreciate your concern. Basically, some cases include any implants, amputation or fracture of femoral shaft which make high artifact on imaging or anatomical failure can occur low accuracy of segmentation. We excluded those cases from our criteria. To advocate for the implementation of the AI-based segmentation model, we provide two supplemental files, the figure of segmentation result image of specific slices of each patient and the running time of the segmentation process.

Major concerns

1. AI is not the main focus of this article or being tested. Consider changing the title as it makes the article seem more about using AI in a novel way instead of an existing segmentation model.

Response: 

Thank you for the feedback. As your recommendation, I changed the title to “Correlation Between Individual Thigh Muscle Volume and Grip Strength In Relation to Sarcopenia with Automated Muscle Segmentation”

2. Patients all have hip fractures, which is associated with sarcopenia. There may not be a true control group to test these associations fully. Would the trends still exist with patients without hip fractures? If so, these associations might be fully helpful in diagnosing sarcopenia.

Response:

Thank you for your valuable feedback. The study utilized data from Real Hip Cohort of hip fracture patients. The analysis was based on CT scans, clinical findings and examinations obtained from these hip fracture patients. Unfortunately, we were unable to acquire CT scans from the general population under the same settings. However, in response to your insightful comment, we plan to include data from the general population by funding for future research projects. Regarding this limitation mentioned, we will incorporate the following statement in the Limitations section.

“Secondly, the study primarily focused on patients with hip fractures, a group inherently linked to sarcopenia, hence lacking a comparative control group without hip fractures to fully evaluate these associations. This limitation, along with an imbalance in our dataset concerning sarcopenia diagnosis because hip fractures are more common in older individuals who are also more likely to have sarcopenia, could potentially lead to biased results[33–35]. 

Further research is warranted to determine whether the observed trends would persist in individuals without hip fractures, as such findings could potentially enhance the utility of these measures in diagnosing sarcopenia. In future studies, it would be useful to examine if the observed trends continue in individuals without hip fractures, as this could help refine the application of these metrics in diagnosing sarcopenia.”

3. Not clear in the first paragraph of the study design if the UNETR model was previously trained in the cited article or trained using the current sample.

Response:

Thank you for your valuable feedback. For any confusion, we modified the sentences.

“By employing the advanced deep learning framework UNETR, tailored for meticulous voxel-level segmentation and sequential data interpretation, we trained the model to achieve notable metrics with 72 CT scan datasets in our previous study. Specifically, our UNETR model attained a Dice score of 0.84 and a relative average volume difference of 0.019%”

4. In the results of the point biserial correlation, the p-values are important but do not show the strength of correlation. What is the correlation value between the muscle groups and sarcopenia? Please report the correlation values in addition to the p-values. Without these, I cannot assess the strength of relationships between sarcopenia and thigh muscles.

Response:

Thank you for the point out of the strength of correlation. We provided strength of correlation in the Table 1, which is not proper expression like correlation. We changed it as Correlation coefficient. Thank you.

5. There are no y-axis labels in Fig 6. 

Response:

Thank you for your feedback. I have revised Figure 6 to make the y-axis.

6. In Fig 6 and Fig 7, change your titles and x-axis to human readable labels instead of variable names.

Response:

Thank you for the feedback. I’ve updated the titles and the x-axis in Figure 6 and 7 to display more intuitive labels.

7. Results section for quantiles of men and women define 3 quartiles instead of 4 as displayed in Fig 6 and 7.

Response: 

Thank you for your insightful feedback. Upon reflection there might be confusion regarding how the quantiles were presented in the Results section. To address this, I have revised the text to all four quantiles clearly. 

“The first quantile (Q1) was at 18kg, the median (Q2) stood at 20kg, the third quantile (Q3) was at 26.3kg and the fourth quantile (Q4) was 37kg. In contrast, the female cohort had a mean grip strength of 12kg, accompanied by a standard deviation of 5.14kg. The first quantile (Q1) was at 5kg, the median (Q2) was at 11.6kg, the third quantile (Q3) was 14.72kg and the fourth quantile (Q4) was 27.6kg.”

8. Fig 6 and 7 are unclear if adjusted or raw muscle volumes are being used for correlations. State this is in the results as well.

Response:

Thank you for your feedback. To set the linearity of skeletal muscle index, We utilized adjusted volume data for Figure 6 and 7. I have clarified the information in the Results section. 

“Subsequent to adjusting for muscle volumes by height (volume/height2), as illustrated in Figure 6, a trend was observed exclusively within the male group, where greater muscle volumes were associated with higher grip strengths.”

9. In results, note the lack of trends among women’s muscle volume and grip strength. This is important finding especially as EWGSOP is recommending muscle performance for sarcopenia diagnosis.

Response:

Thank you for your valuable feedback. As your suggestion, we emphasize the result of the lack of trends among women’s muscle volume and grip strength. The following paragraph is the modified paragraph in the “Results” section.

“As illustrated in Figure 6, a noteworthy trend was observed exclusively within the male group, where greater muscle volumes were associated with higher grip strengths. Furthermore, as shown in Figure 7, the anterior muscle group displayed specific mean volumes across different quartiles: 68.95 in the second quartile (Q2), 93.83 in the third quartile (Q3), and 107.02 in the fourth quartile (Q4). These findings suggest a subtle yet discernible correlation trend between anterior muscle volume and grip strength in male group. This pattern was not observed in the female group, where no significant relationship with muscle volume was found (p > 0.05).”

10. In results, the sentence “These findings suggest a subtle yet discernible correlation between anterior muscle volume and grip strength.” Should be changed to “trend between…” as no correlation was tested.

Response:

Thank you for your correction of the sentence in “Results” section. We modified the sentence like the following sentence.

“

These findings suggest a subtle yet discernible trend between anterior muscle volume and grip strength.

”

11. A lot of the issues of “trends” in Figure 6 and 7 could be tested by comparing the muscle group volumes and ratios to grip strength continuously. You would be able to assess trend strength instead observing possible trends. You could also represent these in tables, which would likely make your results more readable. 

Response:

Thank you for your valuable suggestions on enhancing the statistical presentation. We conducted a Spearman's rank correlation analysis to explore the relationship between grip strength and the volume of each specific thigh muscle. To improve readability and present our findings more effectively, we've incorporated these results into Table 3, titled "Spearman's Rank Correlation between Grip Strength and Individual Thigh Muscle Volume."

12. The last paragraph of the results states there are differences and associations, but none have been tested statistically between grip strength and muscle volumes. This reviewer highly recommends changing this language to “trends” or testing them with Pearson’s correlations and/or ANOVA tests.

Response:

I appreciate your valuable input. I have modified the context of correlation to trends in the paragraphs within the Results section.

13. Please clarify the second statement in the discussion. Is this a correlation between muscle volumes and grip strength or muscle volumes and sarcopenia? If this is reference to muscle volumes and grip strength, that association was not tested as far as this reviewer can tell.

Response:

Thank you for your insights. To avoid any confusion, I have made it clear that this is about the correlation between thigh muscle volumes and sarcopenia, according to the current consensus. Thank you for pointing this out.

“The present study employed grip strength and skeletal muscle index as a consensus marker for diagnosing sarcopenia and found significant correlations between sarcopenia and thigh muscle volumes such as Rectus femoris, Vastus lateralis and Vastus intermedius, particularly in the male cohort. Those specific thigh muscles imply their potential contribution in the onset or progression of sarcopenia in males. Conversely, no significant associations were observed between sarcopenia and each individual muscle volume in female group.”

14. The statement in the discussion- “The quantile analysis further supported these findings showing a clear trend in the male group that higher muscle volumes are correlated with better grip strengths.”- is not supported in the research. No correlation was established statistically.

Response:

Thank you for your guidance. Following your suggestion, I’ve revised the sentence to reflect that we did not conduct a correlation analysis.

“

The analysis of quantile indicates a trend within the male group where increased volumes of the anterior and medial thigh muscles could be linked with stronger grip strength.

”

15. This reviewer would also like to see the point biserial correlations and any of the muscle volume comparisons of the ground truth against the AI segmentation to see how the differences compare to a performance metric (grip strength) and diagnosis of sarcopenia. If this is included, this reviewer feels the title including AI would be more valid.

Response:

Providing an analysis with the ground truth would indeed be valuable. However, we employed a different dataset in Real Hip Cohort for model development which included 60 CT scans for training and 12 CT scans for validation. Creating a ground truth segmentation for a single CT scan requires approximately 4 hours, generating ground truths for all 49 scans would be exceedingly labor-intensive and costly. Consequently, it is difficult to compile a complete set of ground truth data for analysis. We apologize for our inability to present a full ground truth analysis under these conditions.

Introduction:

1. In the introduction, it would be beneficial to include a brief explanation of UNETR and why it is a more accurate choice for assessing individual thigh muscle volumes. This will help to avoid any potential confusion.

Response: 

Thank you for your feedback. I have included a brief explanation of UNETR and previous study to avoid any potential confusion.

“

The UNETR model represents an AI-driven semantic segmentation architecture that merges the U-net and Transformer modes. It excels in accurate voxel-level segmentation of 3D medical images, including CT scans and MRIs. In our previous study, we utilized the UNETR architecture to develop a thigh muscle segmentation model, employing a dataset of 72 entries (60 for training and 12 for validation) from the Real Hip Cohort. Real Hip Cohort includes variety of medical images, such as CT scans, X-rays and Dexa, along with physical performance metrics like grip strength from hip fracture patients.

”

2. The introduction lacks an explanation and reference for why the thigh muscles have a direct impact on grip strength (Ln 65-67). This oversight makes it difficult to understand the primary aim mentioned later (Ln 83-85), particularly the relationship between thigh muscle volumes, grip strength, and sarcopenia, as well as the reason for investigating these factors.

Response: 

Thank you for the feedback. We have updated the paragraph of relationship between thigh muscles and grip strength more logical.

“

Beyond the focus on the hand and forearm muscles in grip strength, other muscle groups like those in the thigh also play an essential role[10]. A loss of balance and stability could potentially compromise the ability to exert force effectively through one’s grip[11]. The gluteus muscles are integral for maintaining balance[12]. Drawing on the insights from these studies, balance-critical muscles, including the lower body and thigh muscles, may have a secondary effect on grip strength by promoting enhanced stability and strength. Based on the findings from these studies, it is hypothesized that lower body muscles, including thos

---

## [Decision Letter · Decision Letter 1]

4 Sep 2024

PONE-D-23-28695R1Correlation Between Individual Thigh Muscle Volume and Grip StrengthIn Relation to Sarcopenia with Automated Muscle SegmentationPLOS ONE

Dear Dr. Yoo,

Thank you for submitting your manuscript to PLOS ONE. After careful consideration, we feel that it has merit but does not fully meet PLOS ONE’s publication criteria as it currently stands. Therefore, we invite you to submit a revised version of the manuscript that addresses the points raised during the review process.

There is really only one thing that has been requested of one Reviewer and this can be done very quickly.

Please identify your code and details of the software environment for your project to the readers of your manuscript. This is consistent with the Journal's goals of data reproducibility (see below). This small revision will increase your data so as to improve reproducibility etc. After receiving this in your revision, i am prepared to accept the manuscript without additional review.

We look forward to receiving your revised manuscript.

Kind regards,

Stephen E Alway, Ph.D.

Academic Editor

PLOS ONE

Journal Requirements:

Reviewers' comments:

Reviewer's Responses to Questions

**Comments to the Author**

1. If the authors have adequately addressed your comments raised in a previous round of review and you feel that this manuscript is now acceptable for publication, you may indicate that here to bypass the “Comments to the Author” section, enter your conflict of interest statement in the “Confidential to Editor” section, and submit your "Accept" recommendation.

Reviewer #2: All comments have been addressed

Reviewer #3: All comments have been addressed

2. Is the manuscript technically sound, and do the data support the conclusions?

Reviewer #2: Yes

Reviewer #3: (No Response)

3. Has the statistical analysis been performed appropriately and rigorously? 

Reviewer #2: Yes

Reviewer #3: Yes

4. Have the authors made all data underlying the findings in their manuscript fully available?

Reviewer #2: Yes

Reviewer #3: Yes

5. Is the manuscript presented in an intelligible fashion and written in standard English?

Reviewer #2: Yes

Reviewer #3: Yes

6. Review Comments to the Author

Reviewer #2: The authors have satisfactorily addressed my comments and concerns. The current version paper is acceptable for publication.

Reviewer #3: The authors may provide open access to their code and details of the software environment to facilitate reproducibility.

7. PLOS authors have the option to publish the peer review history of their article (what does this mean?). If published, this will include your full peer review and any attached files.

Reviewer #2: No

Reviewer #3: No

---

## [Author Response · Author response to Decision Letter 1]

9 Sep 2024

Response: 

Thank you for submitting your manuscript to PLOS ONE. After careful consideration, we feel that it has merit but does not fully meet PLOS ONE’s publication criteria as it currently stands. Therefore, we invite you to submit a revised version of the manuscript that addresses the points raised during the review process.

There is really only one thing that has been requested of one Reviewer and this can be done very quickly.

Please identify your code and details of the software environment for your project to the readers of your manuscript. This is consistent with the Journal's goals of data reproducibility (see below). This small revision will increase your data so as to improve reproducibility etc. After receiving this in your revision, i am prepared to accept the manuscript without additional review.

Response:

Thank you for your valuable feedback. I have specified the software environment we used in the ‘Methods’ section on line 216. Furthermore, the training codes of muscle segmentation model is not publicly available, as the trained model weights and the datasets utilized are not accessible through open access.

“The muscle segmentation model was trained using an Nvidia DGX A100 system, powered by two Nivia A100 Gpus and running on Ubuntu 20.04 operating system. The model was implemented with the Pytorch and MONAI libraries. For the training process, the dice coefficient was utilized as the loss function. The AdamW optimizer was set with a learning rate of 8e-5, weight decay of 1e-5, a batch size of 4 and 3000 epochs. These values were selected through a grid search method, testing different learning rates range from 1e-3 and to 1e-5”

Additionally, the affiliations of the first author, Hyeon Su Kim, and the third author, Hyunbin Kim, have been updated to 'Department of Orthopaedic Surgery, Inha University Hospital, Inha University College of Medicine, Incheon, South Korea.' We have made the changes to reflect this on the manuscript. Thank you for your understanding.

---

## [Editor Report · Decision Letter 2]

1 Oct 2024

Correlation Between Individual Thigh Muscle Volume and Grip Strength

In Relation to Sarcopenia with Automated Muscle Segmentation

PONE-D-23-28695R2

Dear Dr. Yoo,

We’re pleased to inform you that your manuscript has been judged scientifically suitable for publication and will be formally accepted for publication once it meets all outstanding technical requirements.

Kind regards,

Stephen E Alway, Ph.D.

Academic Editor

PLOS ONE

Additional Editor Comments (optional):

Thank you for addressing the concerns of the Reviewers.
---

## [Editor Report · Acceptance letter]

11 Nov 2024

PONE-D-23-28695R2 

PLOS ONE

Dear Dr. Yoo, 

I'm pleased to inform you that your manuscript has been deemed suitable for publication in PLOS ONE. Congratulations! Your manuscript is now being handed over to our production team.

Kind regards, 

on behalf of

Dr. Stephen E Alway 

Academic Editor

PLOS ONE